# Online Billion-Scale Recommender Systems with Macro Graph Neural Networks

## ABSTRACT

Predicting Click-Through Rate (CTR) in billion-scale recommender systems poses a long-standing challenge for Graph Neural Networks (GNNs) due to the overwhelming computational complexity involved in aggregating billions of neighbors. To tackle this, GNN-based CTR models usually sample hundreds of neighbors out of the billions to facilitate efficient online recommendations. However, sampling only a small portion of neighbors results in a severe sampling bias and the failure to encompass the full spectrum of user or item behavioral patterns. To address this challenge, we name the conventional user-item recommendation graph as "micro recommendation graph" and introduce a more suitable **MAcro Recommendation Graph (MAG)** for billion-scale recommendations. MAG resolves the computational complexity problems in the infrastructure by reducing the node count from billions to hundreds. Specifically, MAG groups micro nodes (users and items) with similar behavior patterns to form macro nodes. Subsequently, we introduce tailored **Macro Graph Neural Networks (MacGNN)** to aggregate information on a macro level and revise the embeddings of macro nodes. MacGNN has already served one of the biggest shopping platforms for two months, providing recommendations for over one billion users. Extensive offline experiments on three public benchmark datasets and an industrial dataset present that MacGNN significantly outperforms twelve CTR baselines while remaining computationally efficient. Besides, online A/B tests confirm MacGNN's superiority in billion-scale recommender systems.

## CCS CONCEPTS

• **Information systems** → **Online advertising**; **Web applications**; • **Human-centered computing** → *Social recommendation*.

## KEYWORDS

graph-based CTR prediction, large-scale recommendation

**ACM Reference Format:**
Anonymous Author(s). 2018. Online Billion-Scale Recommender Systems with Macro Graph Neural Networks. In *Proceedings of Make sure to enter the correct conference title from your rights confirmation emai (Conference acronym 'XX)*. ACM, New York, NY, USA, 10 pages. https://doi.org/XXXXXXX.XXXXXXX

## 1 INTRODUCTION

Billion-scale recommender systems, with billions of users, items, and interactions, are prevalent in today's societies [24, 26, 27, 29], such as YouTube [4] and Taobao [16]. At the heart of these billion-scale recommender systems lies Click-Through Rate (CTR) prediction [28]. Its goal is to predict, in real-time, whether a given user will click on a given item. However, due to efficiency requirements, while Graph Neural Networks (GNNs) have shown significant performance in collaborative filtering recommendation tasks [13, 23], they are not well-suited for CTR tasks. This is because performing graph neural networks over billion-scale neighbors leads to overwhelming computational complexity. It is crucial to develop appropriate graph neural networks capable of handling recommender systems with billions of users, items, and interactions.

Existing GNN models typically create the graph by linking users to their interacted (clicked) items. In this scenario, if a user interacts with a highly popular item with billions of interactions, then the subgraph of that user will potentially have billions of 2-hop neighbors. To reduce computational complexity, PinSage [26] randomly selects a fixed number of 1-hop and 2-hop neighbors for both users and items. GLSM [20] and GMT [15] introduce importance-based and similarity-based scoring mechanisms to filter the most suitable hundreds of 1-hop and 2-hop neighbors. Besides traditional CTR models introduce filtering strategies to accelerate the inferring process. DIN [31] and DIEN [30], typically truncate a user's recently 150 interacted items. SIM [17] introduced a search-based strategy to filter the most relevant items from the user's entire historical behavior. However, traditional CTR models fail to consider the filtering for the subgraph of items or the 2-hop neighbors of users.

Though the above strategies can reduce the neighbor size for GNNs, these approaches still face the following limitations in billion-scale recommender systems.

**1. Severe Sampling Bias:** In Figure 1(a), we illustrate the distribution of neighbor numbers in the user-item clicking interaction graph within a real-world shopping platform. Both users and items exhibit a substantial number of 1-hop and 2-hop neighbors. Sampling only a few hundred neighbors can only cover about 5% of user 1-hop neighbors and 0.2% of item 1-hop neighbors. Sampling such small portions cannot accurately represent the entire spectrum of neighbors and may lead to severe sampling bias.

**2. Unfitted Users/Items Sampling:** As shown in Figure 1(a), users exhibit vastly different number distributions compared to items. For example, users have significantly more 2-hop neighbors and significantly fewer 1-hop neighbors than items. It is inappropriate to sample users and items using the same approach.

**3. Ambiguous Neighbor Counts:** The sampled neighbors do not accurately represent the true number of interactions prior to the sampling process for users and items. For instance, a user with hundreds of historical interactions will yield the same sample size as another user with millions of historical interactions.

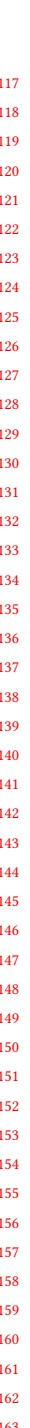

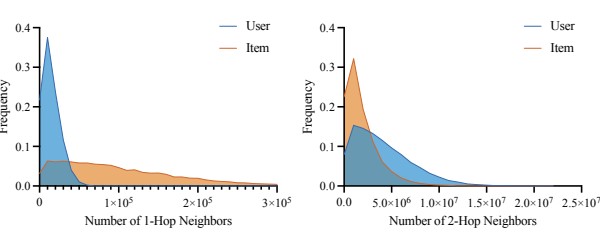

(a) Neighborhood number distribution of micro graphs.

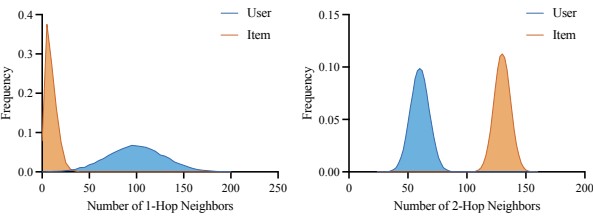

(b) Neighborhood number distribution of macro graphs.

**Figure 1: Illustration of neighbor number distributions in micro and macro user-item clicking interaction graphs within a real-world billion-scale recommender system.**

The main problem behind the mentioned issues arises from relying on sampling strategies to decrease the size of neighbors. Instead, it's more promising to boost the expressive capacity of graph nodes and significantly reduce the neighbor size by grouping nodes into macro nodes. This grouping approach allows models to overcome the inherent limitations of sampling strategies by eliminating the need for the sampling process entirely. However, actualizing a grouping strategy for recommendation graphs introduces the following challenges.

**1. Grouping Strategy:** Identifying an optimal grouping strategy for user and item nodes into macro nodes is non-trivial, as it demands a careful balance between reducing complexity and maintaining the integrity of original behavioral patterns.

**2. Subgraph Definition:** Constructing edges between macro nodes is complex due to the necessity of representing aggregated interactions between their constituent user/item nodes accurately. Additionally, defining the subgraph for a given user/item using macro nodes demands innovative approaches.

**3. Recommending with Macro Nodes:** Each macro node represents a group of user/item nodes, and the edge between two macro nodes signifies the connections between two groups of nodes. It is challenging to extract the behavioral pattern of a user/item node based on its newly constructed macro-node subgraphs.

By addressing the three challenges mentioned above, we propose a more suitable **MA**cro Recommendation **G**raph (MAG) for billion-scale recommendations. MAG groups user/item nodes based on similar behaviors to create macro nodes, as illustrated in Figure 2. This grouping reduces the number of neighbors from billions to hundreds. As depicted in Figure 1(b), MAG now only consists of hundreds of 1-hop and 2-hop neighbors. This reduction allows billion-scale recommender systems to alleviate the adverse consequences of sampling only a small portion of neighbors.

To achieve this, we introduce tailored Macro Graph Neural Networks (MacGNN) to aggregate the macro information for the target user/item with our specially designed MAG, facilitating accurate and efficient click-through rate prediction for online billion-scale recommender systems. Our paper's primary contributions can be summarized as follows:

- We create a customized macro recommendation graph, which involves constructing the macro node, macro edge, and macro subgraph. This helps reduce the neighbor size from billions to hundreds, making it easier for GNNs to operate in online billion-scale recommender systems.
- We propose a novel macro-scale recommendation paradigm known as the Macro Graph Neural Network (MacGNN). This framework efficiently aggregates macro-graph information and updates macro-node embeddings to enable online click-through rate prediction for billion-scale recommender systems.[1]
- MacGNN has been serving a major shopping platform for two months, offering recommendations to more than one billion users. Additionally, we introduce our online implementation to enable online updates of macro nodes and macro edges.
- Extensive offline experiments conducted on three public benchmark datasets and a billion-scale industrial dataset demonstrate that MacGNN outperforms twelve state-of-the-art CTR baselines while maintaining competitive efficiency. Furthermore, online A/B tests have confirmed the superiority of MacGNN in real-world billion-scale recommender systems.

## 2 PRELIMINARIES

In this section, we first present the basic notations in CTR prediction. Then, we present the concept of micro nodes, micro edges, and micro recommendation graphs for recommender systems. Finally, we introduce the definition of our macro recommendation graph.

***CTR Prediction.*** Supposed the set of users and items as $\mathcal{U} = \{u_1, ..., u_n\}$, and $\mathcal{I} = \{i_1, ..., i_m\}$, respectively, where $|\mathcal{U}| = n$ and $|\mathcal{I}| = m$ denotes the number of users and items. In real-world recommender systems, CTR models correspond to a click or not problem. When item $i$ is exposed to user $u$, user $u$ will have two reflections: (i) having a positive behavior toward the item $i$ such as click or purchase, or (ii) having a negative behavior toward the item $i$ such as neglect or dislike. Thus, given the target user-item pair as $(u, i)$, the corresponding interaction $y_{ui}$ can be present as:

$$y_{ui} = \begin{cases} 1, & \text{if } u \text{ exhibits positive behavior towards } i; \\ 0, & \text{if } u \text{ exhibits negative behavior towards } i. \end{cases} \quad (1)$$

Given a target user-item pair $(u, i)$, the CTR prediction task is to predict the target user $u$'s positive behavior probability $\hat{y}_{ui}$ on target item $i$. In form, the aim of a CTR model is to learn an accurate prediction function $\mathcal{F}(\cdot)$, namely the predicted clicking probability $\hat{y}_{ui} = \mathcal{F}(u, i)$, to minimize the difference from $\hat{y}_{ui}$ to $y_{ui}$.

***Micro Node.*** Starting with several popular works [13, 23, 26], GNN-based recommendation models usually connect users with their interacted (e.g. clicked or purchased) items. Under this setting, users and items are treated as micro nodes. Specifically, each user $u$ and item $i$ is associated with a trainable embedding $E_u \in \mathbb{R}^d$ and $E_i \in \mathbb{R}^d$, where $d$ is the embedding dimension size.

---

[1]Source code is available at https://anonymous.4open.science/r/MacGNN.

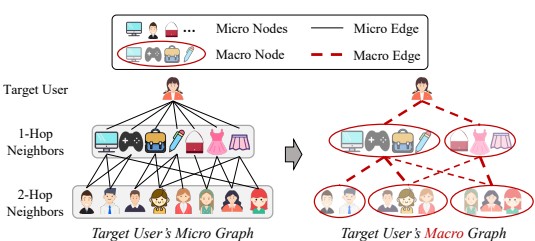

**Figure 2: Sketch map of the construction of the macro graph.**

***Micro Edge.*** As stated in Eq. (1), the user-item behaviors actually provide the most raw material for edges. Given the micro user-item interaction matrix $\mathcal{R} \in \mathbb{R}^{|\mathcal{U}| \times |\mathcal{I}|}$, where $|\mathcal{R}|$ is the total number of interactions. Each element $r_{ui} \in \mathcal{R}$ reflects whether users $u$ have a positive interaction with item $i$, namely $r_{ui} = y_{ui}$.

***MIcro Recommendation Graph (MIG).*** After defining the micro nodes and micro edges, the MIG can be represented as $\mathcal{G} = (\mathcal{U}, \mathcal{I}, \mathcal{R})$. For user interest models, the user behavior sequences can be given as the first-order neighbor of the user $u$ as $\mathcal{N}_u^{(1)}$, where $\mathcal{N}_u^{(k)}$ denotes the $k^{th}$-hop neighbors of user $u$.

According to the definition of MIG, when GNNs predict the CTR of a given user-item pair $(u, i)$, GNNs first construct the micro subgraph of the target user/item and then extract the embeddings according to MIG. When the graph size grows to a billion-scale, the subgraph may contain billions of micro nodes, which means only loading the embeddings of the subgraph is difficult to accomplish.

***MAcro Recommendation Graph (MAG).*** Our proposed MAG can be defined as $\widetilde{\mathcal{G}} = (\widetilde{\mathcal{U}}, \widetilde{\mathcal{I}}, \widetilde{\mathcal{R}})$, where $\widetilde{\mathcal{U}}, \widetilde{\mathcal{I}}$, and $\widetilde{\mathcal{R}}$ are the macro user nodes, macro item nodes, and macro edges respectively, and $\widetilde{\mathcal{N}}_v^{(k)}$ represents the $k^{th}$-hop macro neighbors of node $v$. Specifically, each macro node $v$ is associated with a trainable embedding $\widetilde{E}_v \in \mathbb{R}^d$. With MAG, MacGNN only needs to aggregate hundreds of macro nodes, significantly reducing computational complexity.

## 3 METHODOLOGY

In this section, we first formally introduce the concept of Macro Recommendation Graphs and introduce how to design macro nodes and macro edges. Then we present the macro graph neural network for CTR prediction. Finally, we illustrate the implementation architecture of our real-world billion-scale recommender system.

### 3.1 Macro Recommendation Graph (MAG)

*3.1.1* ***Constructing Macro Nodes.*** As presented in the preliminaries, MIG records the detailed micro node and micro edge for each user and item. Then, given any user or item, the GNNs have to access the embeddings of each hop of micro nodes to infer the behavior pattern of the given user or item, which is computationally inconvenient and raises responsible delays. Motivated by this, MAG presents the behavior pattern within macro nodes rather than listing all the micro nodes and utilizes the GNNs to extract the behavior pattern from detailed micro nodes.

Intuitively, the macro nodes are designed to represent the behavior pattern of a set of micro nodes, while all the micro nodes inside share similar behavior patterns. Thus, we conduct the behavior pattern grouping to map the micro nodes into specific macro behavior

nodes, with the objection of minimizing the behavioral pattern gap between macro nodes assigned to the same macro node [10].

Specifically, given the micro user-item interaction matrix $\mathcal{R} \in \mathbb{R}^{|\mathcal{U}| \times |\mathcal{I}|}$, for a given user/item micro node $v$, we first obtain its behavior embedding $\boldsymbol{b}_v$ as follows:

$$\boldsymbol{b}_v = ||[\boldsymbol{R}]_v||_2^2 = ||\boldsymbol{r}_v||_2^2, \qquad \boldsymbol{R} = \begin{cases} \mathcal{R}, & v \in \mathcal{U}; \\ \mathcal{R}^\top, & v \in \mathcal{I}. \end{cases} \qquad (2)$$

where $|| \cdot ||_2^2$ is the $L_2$ norm. Then, to obtain each macro node $C_k$, we conduct the behavior pattern grouping based on behavior embeddings of micro nodes. Specifically, we first randomly initialize $K$ macro centroids $\{\boldsymbol{\mu}_1, ..., \boldsymbol{\mu}_k, ..., \boldsymbol{\mu}_K\}$, where $\boldsymbol{\mu}_k \in \mathbb{R}^d$ is the centroids of macro node $C_k$, $K \ll n$ and $m$ is the hyperparameter set as the macro node number, and we denote $K$ for macro user node and macro item node is $\widetilde{n}$ and $\widetilde{m}$, respectively. Then, we explore and assign micro nodes to the appropriate macro node based on their behavior patterns, and update the centroid of macro nodes iteratively. The process can be expressed as:

$$\boldsymbol{\mu}_k = \frac{1}{|C_k|} \sum_{x_v=k, \boldsymbol{b}_v \in C_k} \boldsymbol{b}_v, \qquad (3)$$

where $|C_k|$ is the number of micro nodes within $C_k$, and $x_v$ is the macro node index that $v$ is assigned to. Further, the optimization objection of the behavior pattern grouping is:

$$\min_{\substack{x_1,...,x_{m+n} \\ \boldsymbol{\mu}_1,...,\boldsymbol{\mu}_K}} J(x_1, ..., x_{m+n}; \boldsymbol{\mu}_1, .., \boldsymbol{\mu}_K)$$
$$\triangleq \sum_{k=1}^{K} \sum_{x_v=k, \boldsymbol{b}_v \in C_k} \sqrt{(\boldsymbol{b}_v - \boldsymbol{\mu}_k)(\boldsymbol{b}_v - \boldsymbol{\mu}_k)^\top}. \qquad (4)$$

where $J$ is the objection function of behavior pattern grouping. As shown in Figure 2, the micro nodes with similar behavior patterns will be composed of a macro node. Note that each macro node $v$ will also be assigned a trainable embedding $\widetilde{E}_v \in \mathbb{R}^d$.

*3.1.2* ***Organizing Macro Edges.*** Macro edges depict relationships between two macro nodes within a specific user/item subgraph, signifying the behavioral patterns within that subgraph. It's important to note that macro edges have a distinct design compared to micro edges. The micro edges present connections between fixed micro user nodes and micro item nodes. Since micro nodes remain constant, the micro edges are also fixed. In contrast, macro edges capture the connection strength between two macro nodes in a subgraph, which is tailored to each user and item subgraph.

In Figure 2, the user $v$ is depicted as having two 1-hop macro nodes, each with macro edge weights of 4 and 3, respectively. Moving to the second hop, the user extends to three macro nodes, and these macro edges represent the connections between the 1-hop macro nodes and the 2-hop macro node. Formally, we use $\widetilde{C} = \{C_1, C_2, \ldots, C_{\widetilde{n}+\widetilde{m}}\}$ to represent the entire set of macro nodes in the MAG. We employ $\widetilde{\mathcal{R}}_{v;p,q}^{(k)}$ to denote the macro edge for any user/item node $v$ with its $k^{th}$-hop neighbors, where $C_{v;p}^{(k-1)}$ represents the macro node in $(k-1)^{th}$-hop macro neighbors $\widetilde{\mathcal{N}}_v^{(k-1)}$, and $C_{v;q}^{(k)}$ represents the macro node in $k^{th}$-hop macro neighbors

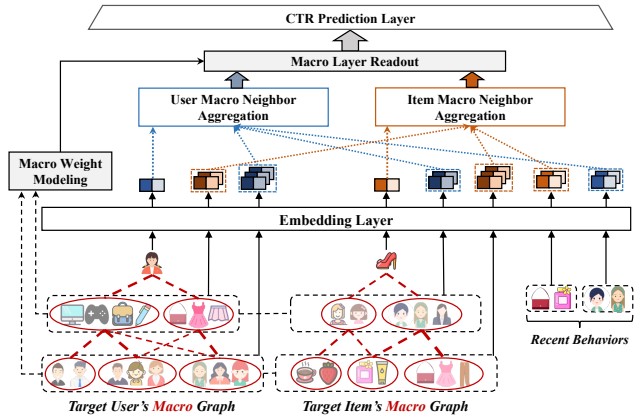

**Figure 3: The model architecture of the proposed MacGNN.**

$\widetilde{\mathcal{N}}_v^{(k)}$. Thus the weight of macro edges can be computed as:

$$\widetilde{\mathcal{R}}_{v;p,q}^{(k)} = \sum_{a \in C_{v;p}^{(k-1)}, b \in C_{v;q}^{(k)}} r_{ab}, \tag{5}$$

where $C_{v;p}^{(k-1)} = C_{v;p} \cap \mathcal{N}_v^{(k-1)}$ represents the macro nodes related to node $v$ within its $(k-1)^{th}$-hop neighbors and $C_{v;q}^{(k)} = C_{v;q} \cap \mathcal{N}_v^{(k)}$ represents the macro nodes related to node $v$ within its $k^{th}$-hop neighbors. In § 3.3, we will introduce how to get online updating macro edges on billion-scale recommender systems. Finally, after transforming micro recommendation graphs into macro recommendation graphs, MAGs have significantly fewer nodes and edges by extracting behavior patterns explicitly into macro nodes.

## 3.2 Macro Graph Neural Network

*3.2.1 **Macro Weight Modeling**.* The overall framework of our proposed MacGNN is shown in Figure 3. To better identify the target user/item preferences over a certain macro node, we design the macro weight modeling for macro neighbors according to the weights of connected macro edges.

In order to avoid the excessive gap between the macro edge weights of hot nodes and cold nodes and conduct modeling flexibly, we equip the macro weight modeling with logarithmic smoothing and temperature-based softmax activation. Formally, take the target user/item $v$ as an example, given a macro node $q$ in its $k^{th}$-hop neighborhood, the macro weight $w_{v;q}^{(k)}$ of $q$ toward the target user/item $v$ is calculated as:

$$s_{v;q}^{(k)} = \log\left(\sum_{p \in \widetilde{\mathcal{N}}_v^{(k-1)}} \widetilde{\mathcal{R}}_{v;p,q}^{(k)} + 1\right), \quad w_{v;q}^{(k)} = \frac{exp\left(s_{v;q}^{(k)}/\tau\right)}{\sum_{j \in \widetilde{\mathcal{N}}_v^{(k)}} exp\left(s_{v;j}^{(k)}/\tau\right)}, \tag{6}$$

where $\tau$ is a temperature coefficient hyper-parameter [1]. These modeled weights represent the importance of these macro neighboring nodes in the target user/item's historical interactions.

*3.2.2 **Marco Neighbor Aggregation & Layer Readout**.* To mine the macro relationships effectively and efficiently, we first design a macro neighbor aggregation architecture rather than a time-consuming recursive graph convolution. Then, we propose

the macro layer readout to aggregate the macro information of the target user and item.

**Macro Neighbor Aggregation.** Due to the different semantics of users and items, we utilized two separate macro neighbor aggregation modules without parameter sharing for user-type macro nodes and item-type macro nodes, respectively.

For user-type target nodes and their $k^{th}$-hop user-type macro neighbors, the aggregation function $MNA_u$ can be defined as:

$$MNA_u(u, p \in \widetilde{\mathcal{N}}_u^{(k)}, E_u, \widetilde{E}_p, \widetilde{\mathcal{N}}_u^{(k)}; Q_u, K_u, V_u)$$
$$\triangleq \sum_{p \in \widetilde{\mathcal{N}}_u^{(k)}} \sigma\left(\langle Q_u \cdot \widetilde{E}_p, K_u \cdot E_u \rangle\right) \cdot V_u \cdot \widetilde{E}_p, \tag{7}$$

where $Q_u, K_u, V_u \in \mathbb{R}^{d \times d'}$ are trainable self-attention matrics for user-type nodes, $\langle \cdot \rangle$ is the inner product function, and $\sigma(\cdot)$ is the softmax activation function. Specifically, given the given target user $u$ and a user-type macro node $p$ in its $k^{th}$-hop neighborhood (such as the node in target user $u$'s 2-hop macro neighborhood and target item $i$'s 1-hop macro neighborhood), the process is expressed as:

$$\alpha_{u,p} = \frac{exp\left((Q_u \cdot \widetilde{E}_p)(K_u \cdot E_u)^\top/\sqrt{d}\right)}{\sum_{j \in \widetilde{\mathcal{N}}_u^{(k)}} exp\left((Q_u \cdot \widetilde{E}_j)(K_u \cdot E_u)^\top/\sqrt{d}\right)}, \tag{8}$$

$$\widetilde{Z}_{u,p} = \alpha_{u,p} \cdot (V_u \cdot \widetilde{E}_p), \tag{9}$$

where $\widetilde{Z}_{u,p}$ is the aggregated macro embedding. Similarly, for the item-type target node $i$ and its macro item-type neighbor $p$ in the $k^{th}$-hop neighborhood, the aggregation function $MNA_i$ to obtain the aggregated macro embedding $\widetilde{Z}_{i,q}$ can be derived in similar ways using separating parameters as:

$$MNA_i(i, q \in \widetilde{\mathcal{N}}_i^{(k)}, E_i, \widetilde{E}_q, \widetilde{\mathcal{N}}_i^{(k)}; Q_i, K_i, V_i)$$
$$\triangleq \sum_{q \in \widetilde{\mathcal{N}}_i^{(k)}} \sigma\left(\langle Q_i \cdot \widetilde{E}_q, K_i \cdot E_i \rangle\right) \cdot V_i \cdot \widetilde{E}_q, \tag{10}$$

where $Q_i, K_i, V_i \in \mathbb{R}^{d \times d'}$ are trainable self-attention matrics for item-type nodes.

**Macro Layer Readout.** With the co-consideration of macro weight modeling and macro neighbor aggregation, we can measure the importance of the specific neighboring macro node from different perspectives. Thus, the representation of a specific-hop macro neighborhood of the target user/item node can be obtained by the following layer readout:

$$E_u^{(l_u)} = \sum_{j \in \widetilde{\mathcal{N}}_u^{(l_u)}} w_{u,j} \cdot \widetilde{Z}_{u,j}, \quad E_i^{(l_i)} = \sum_{j \in \widetilde{\mathcal{N}}_i^{(l_i)}} w_{i,j} \cdot \widetilde{Z}_{i,j}, \tag{11}$$

where $E_u^{(l_u)}$ and $E_i^{(l_i)}$ denote the $l_u$-hop/$l_i$-hop readout representation of target user/item, respectively.

*3.2.3 **Recent Behavior Modeling**.* The above macro modeling takes into account the general and stable behavioral characteristics of the target node. Leveraging the learned knowledge at such a macro level, we further consider the information of recent behavior to better extract users' changing short-term interests and the evolving interaction patterns of items [5, 20].

Formally, for the target user $u$ and target item $i$, the few most recently interacted neighbor sequence $RS_u$ and $RS_i$ are utilized and

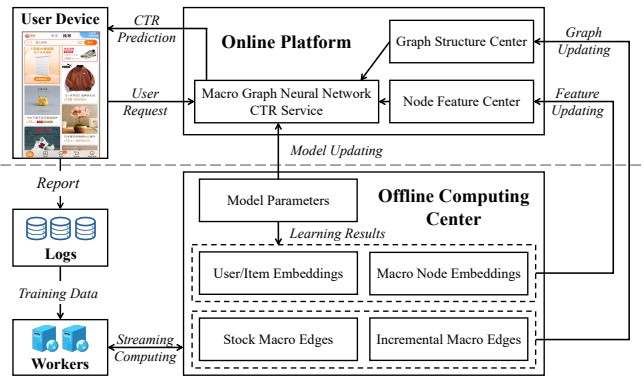

**Figure 4: The system architecture for online deployment.**

their embeddings are co-trained with the macro nodes in the above aggregation functions, respectively.

$$Z_{u,rs_p} = MNA_i(i, rs_p \in RS_u, E_i, E_{rs_p}, RS_u; Q_i, K_i, V_i),$$
$$Z_{i,rs_q} = MNA_u(u, rs_q \in RS_i, E_u, E_{rs_q}, RS_i; Q_u, K_u, V_u), \quad (12)$$

$$E_i^{rs} = \sum_{rs_q \in RS_i} Z_{i,rs_q}, \quad E_u^{rs} = \sum_{rs_p \in RS_u} Z_{u,rs_p}, \quad (13)$$

where $E_u^{rs}$ and $E_i^{rs}$ are the representation of the few macro node sequence $RS_u$ and $RS_i$. The sequence length of the few recent behaviors for auxiliary training is set to 20. Note that the number of recent nodes for modeling is much smaller than the hundreds of sequence lengths in the advanced interest models [30, 31].

*3.2.4* **CTR Prediction Layer**. With the obtained informative representations, we utilize them for the final CTR prediction for the target user $u$ and target item $i$ as the following calculation:

$$\hat{y}_{u,i} = MLP\left( (\|_{l_u}^K E_u^{(l_u)}) \parallel (\|_{l_i}^K E_i^{(l_i)}) \parallel E_u^{rs} \parallel E_i^{rs} \parallel E_u \parallel E_i \right), \quad (14)$$

where the architecture and parameter settings of the MLP are the same as previous works [30, 31].

To train and optimize the model parameters, we apply the binary cross-entropy loss as the model objective function. Formally, for each user-item pair $(u, i)$ in training set $\mathcal{TS}$, the adopted objective function can be expressed as:

$$\mathcal{L}_{bce} = -\frac{1}{|\mathcal{TS}|} \sum_{(u,i) \in \mathcal{TS}} y_{u,i} \log(\hat{y}_{u,i}) + (1-y_{u,i}) \log(1-\hat{y}_{u,i}), \quad (15)$$

where $\hat{y}_{u,i}$ is the predicted CTR and $y_{u,i}$ is the ground-truth label. Then, the overall objective function of MacGNN is as follows:

$$\mathcal{L} = \mathcal{L}_{bce} + \lambda \cdot \|\boldsymbol{\theta}\|_2^2, \quad (16)$$

where $\lambda \cdot \|\boldsymbol{\theta}\|_2^2$ denotes the $L_2$ regularization to avoid over-fitting.

## 3.3 Online Implementation

In this section, we present the online deployment of MacGNN on a leading e-commerce platform's homepage. MacGNN has provided stable and precise recommendations to over 1 billion users and 2 billion items, analyzing more than 12 trillion interactions since August 2023.

The core architecture to implement the proposed MacGNN model is presented in Fig. 4, including the workflow of both offline computing and online serving. Offline computing can compute the necessary embeddings and graph structures without affecting the online service. Specifically, offline computing is based on a distributed machine learning platform, which loads log data to train the model parameters and embeddings. Then the learned user/item embedding and the macro node embedding are uploaded to the graph feature center for online serving.

Another job of offline computing is the graph structure updates. For example, during shopping events like Black Friday or Singles' Day, certain popular items can receive billions of clicks within seconds. In such scenarios, we employ two modules to facilitate graph structure updates. The stock micro edges are computed offline on a daily basis (or even hourly if necessary). Meanwhile, the incremental micro edges store the micro edges generated in real-time. Since the macro edge weights (Eq. (5)) are defined through summation, the complete micro edge weights can be computed by adding the stock macro edge weights and the incremental macro edge weights.

With the help of offline computing, during the online inferring process, MacGNN can directly get the macro edges through the graph structure center and get the macro node embeddings through the graph feature center. Since MacGNN only considers the macro node, we can give the upper bound of the related node number as $O((\tilde{n} + \tilde{m}))$. On the contrary, the expected related node number of traditional micro GNNs can be given as $O(\frac{|\mathcal{R}|^2}{m \times n})$. Specifically, we construct 200 macro nodes for users and 300 macro nodes for items. Then the micro GNNs will consider about 6 million times more nodes of the MacGNN if micro GNNs consider all the micro nodes in the billion-scale recommender system.

## 4 EXPERIMENTS

In this section, we conduct comprehensive experiments on both offline datasets and real-world online recommendation systems, aiming to answer the following research questions. **RQ1:** How does MacGNN perform compared to state-of-the-art models? **RQ2:** How efficient is the proposed MacGNN? **RQ3:** What is the effect of different components in MacGNN? **RQ4:** How do key hyper-parameters impact the performance of MacGNN? **RQ5:** How does MacGNN perform on billion-scale real-world recommendation platforms?

## 4.1 Experimental Setup

*4.1.1 Datasets.* We conduct comprehensive experiments on three widely used benchmark datasets **MovieLens** [11], **Electronics** [14], and **Kuaishou** [7], and one large-scale industrial dataset from **one of the biggest shopping platforms** to verify the effectiveness of MacGNN. The statistics of these datasets are shown in Table 1. The detailed description of these datasets is illustrated in Appendix A.

**Table 1: Statistics of the experimental datasets.**

| Dataset | # Users | # Items | # Interactions | # Categories |
|---|---|---|---|---|
| MovieLens | 71,567 | 10,681 | 10,000,054 | 21 |
| Electronics | 192,403 | 63,001 | 1,689,188 | 801 |
| Kuaishou | 7,176 | 10,728 | 12,530,806 | 31 |
| Industrial | 170,000,000 | 310,000,000 | 118,000,000,000 | 27,452 |

**Table 2: CTR prediction comparison results over *five* trial runs (↑: the higher, the better; ↓: the lower, the better). The best baseline(s) are highlighted with underlining.**

| Model | MovieLens | | | Electronics | | | Kuaishou | | |
|---|---|---|---|---|---|---|---|---|---|
| | AUC (↑) | GAUC (↑) | Logloss (↓) | AUC (↑) | GAUC (↑) | Logloss (↓) | AUC (↑) | GAUC (↑) | Logloss (↓) |
| Wide&Deep | 0.7237±0.0008 | 0.6922±0.0009 | 0.6072±0.0020 | 0.8242±0.0009 | 0.8247±0.0008 | 0.5132±0.0033 | 0.8202±0.0023 | 0.7761±0.0006 | 0.4922±0.0025 |
| DeepFM | 0.7215±0.0015 | 0.6910±0.0011 | 0.6080±0.0026 | 0.8064±0.0028 | 0.8066±0.0028 | 0.5352±0.0081 | 0.8207±0.0014 | 0.7753±0.0007 | 0.4922±0.0023 |
| AFM | 0.7199±0.0008 | 0.6884±0.0007 | 0.6091±0.0013 | 0.7995±0.0008 | 0.7999±0.0009 | 0.5330±0.0008 | 0.8184±0.0034 | 0.7731±0.0049 | 0.4969±0.0041 |
| NFM | 0.7156±0.0039 | 0.6850±0.0042 | 0.6171±0.0078 | 0.8044±0.0009 | 0.8049±0.0009 | 0.5372±0.0033 | 0.8186±0.0045 | 0.7717±0.0022 | 0.4951±0.0040 |
| DIN | 0.7248±0.0010 | 0.6974±0.0005 | 0.6143±0.0043 | 0.8295±0.0026 | 0.8307±0.0030 | 0.5186±0.0028 | 0.8208±0.0019 | 0.7792±0.0005 | 0.4978±0.0031 |
| DIEN | 0.7262±0.0010 | 0.6958±0.0009 | 0.6112±0.0020 | 0.8313±0.0031 | 0.8323±0.0027 | 0.5167±0.0056 | 0.8273±0.0016 | 0.7783±0.0009 | 0.4943±0.0054 |
| UBR4CTR | 0.7245±0.0002 | 0.6943±0.0010 | 0.6233±0.0076 | 0.8300±0.0005 | 0.8299±0.0006 | 0.5056±0.0007 | 0.8266±0.0005 | 0.7799±0.0006 | 0.4907±0.0020 |
| SIM | 0.7255±0.0014 | 0.6950±0.0012 | 0.6254±0.0094 | 0.8296±0.0033 | 0.8305±0.0031 | 0.5186±0.0062 | 0.8273±0.0005 | 0.7800±0.0005 | 0.4906±0.0021 |
| PinSage | 0.7298±0.0017 | 0.7069±0.0017 | 0.6121±0.0039 | 0.8136±0.0027 | 0.8133±0.0027 | 0.5269±0.0078 | 0.8163±0.0019 | 0.7810±0.0006 | 0.5037±0.0041 |
| LightGCN | 0.7305±0.0009 | 0.7077±0.0012 | 0.6122±0.0061 | 0.8329±0.0011 | 0.8333±0.0010 | 0.5101±0.0049 | 0.8139±0.0019 | 0.7803±0.0014 | 0.5068±0.0041 |
| GLSM | 0.7320±0.0003 | 0.7096±0.0007 | 0.6088±0.0035 | 0.8318±0.0026 | 0.8324±0.0026 | 0.5112±0.0066 | 0.8170±0.0012 | 0.7811±0.0004 | 0.5031±0.0059 |
| GMT | 0.7353±0.0014 | 0.7097±0.0010 | 0.6003±0.0023 | 0.8313±0.0020 | 0.8322±0.0024 | 0.5110±0.0083 | 0.8215±0.0018 | 0.7803±0.0017 | 0.4981±0.0020 |
| **MacGNN** | **0.7458±0.0006** | **0.7198±0.0007** | **0.5886±0.0027** | **0.8444±0.0009** | **0.8458±0.0008** | **0.4892±0.0040** | **0.8306±0.0013** | **0.7813±0.0010** | **0.4872±0.0026** |

*4.1.2 Competitors.* To evaluate the effectiveness of MacGNN, we compare it with twelve representative state-of-the-art CTR prediction models into three main groups. (i) *Feature Interaction-based Methods*: **Wide&Deep** [2], **DeepFM** [9], **AFM** [25], and **NFM** [12]. (ii) *User Interest-based Methods*: **DIN** [31], **DIEN** [30], **UBR4CTR** [18], and **SIM** [17]. (iii) *Graph-based Methods*: **PinSage** [26], **Light-GCN** [13], **GLSM** [20], and **GMT** [15]. We leave the details of these baseline models in Appendix B.

*4.1.3 Hyperparameter Setting.* For all models, the embedding size is fixed to 10 and the embedding parameters are initialized with the Xavier method [8]. Shapes of the final MLP for all models are set to [200, 80, 2] as previous works [30, 31]. The learning rate of MacGNN is searched from $\{1 \times 10^{-2}, 5 \times 10^{-3}, 1 \times 10^{-3}\}$, the regularization term is searched from $\{1 \times 10^{-4}, 5 \times 10^{-5}, 1 \times 10^{-5}\}$. The batch size is set to 1024 for all models and the Adam optimizer is used.

*4.1.4 Evaluation Metrics.* We evaluate the models with three widely-adopted CTR prediction metrics including AUC [6], GAUC [31], and Logloss [32]. The *higher* AUC and GAUC value indicates higher CTR prediction performance, and the *lower* Logloss value indicates higher CTR prediction performance. Note that we run all the experiments *five* times with different random seeds and report the average results with standard deviation to prevent extreme cases.

## 4.2 Offline Evaluation (RQ1)

In this subsection, we compare our proposed MacGNN with twelve state-of-the-art baseline models on the four experimental datasets. The comparison results on the AUC and GAUC metrics are reported in Table 2 and Table 3, with the following observations:

**MacGNN can achieve significant improvements over state-of-the-art methods on all experimental datasets.** From the tables, we observe that the proposed MacGNN achieves the highest AUC and GAUC performance and the lowest Logloss results. Specifically, for the Logloss metric, MacGNN outperforms the best baseline by 1.95%, 4.10%, 0.71%, and 0.93% on MovieLens, Electronics, Kuaishou, and the industrial dataset, respectively. For all the AUC, GAUC, and Logloss metrics, MacGNN brings effective gains

of 1.00%, 0.93%, and 1.70% on average respectively. These comparison results verify that taking into account the graph information in a macro perspective of MacGNN contributes to achieving better interest modeling and CTR prediction performance.

**The graph-based methods perform relatively well than other types of baseline models.** Comparing the three main categories of baseline models, we can find the graph-based models (i.e. PinSage, LightGCN, GLSM, and GMT) obtain relatively better results than user interest modeling and feature interaction methods, which indicates that apart from the directly interacted neighborhood, incorporating high-order graph information can reflect the useful implicit preferences of the target user-item pair, and is significant for the overall CTR prediction performance.

**Increasing the modeling range through node sampling does not necessarily bring effective gains in all scenarios.** These results show that applying node sampling-based methods (e.g. UBR4CTR, SIM, and GLSM) to consider behaviors does not consistently bring improvements to the performance. This suggests that modeling node interests by only searching and sampling similar nodes based on certain rules may not be accurate enough. Additionally, retrieving neighbors beyond the 1-hop using GLSM resulted in relatively better performance compared to UBR4CTR and SIM, also indicating that the higher-order interaction information is meaningful. The designed macro graph paradigm of MacGNN avoids this issue, which is an important factor contributing to its optimal performance.

## 4.3 Efficiency Study (RQ2)

Since CTR prediction has to infer the user's intent in real-time and thus the computational efficiency of models is also an important evaluation factor [22]. Hence, to verify the efficiency of MacGNN, we compare the average response time per user-item pair between MacGNN and the well-performed and representative baselines: feature interaction-based model **Wide&Deep**, user interest-based model **DIN** and the node searching scheme **SIM**, graph-based recursively convolution method **LightGCN** and graph transformer-based method **GMT**. Note that we present the online inference time on real-world recommender systems.

**Table 3: Comparison results on the industrial dataset.**

| Industrial | AUC (↑) | GAUC (↑) | Logloss (↓) |
|---|---|---|---|
| Wide&deep | 0.8123±0.0021 | 0.6908±0.0024 | 0.5223±0.0009 |
| DeepFM | 0.8169±0.0012 | 0.6982±0.0036 | 0.5202±0.0018 |
| AFM | 0.8103±0.0008 | 0.6866±0.0021 | 0.5301±0.0020 |
| NFM | 0.8112±0.0031 | 0.6823±0.0043 | 0.5286±0.0032 |
| DIN | 0.8225±0.0017 | 0.6963±0.0013 | 0.5022±0.0012 |
| DIEN | 0.8231±0.0042 | 0.7008±0.0018 | 0.5009±0.0021 |
| UBR4CTR | 0.8263±0.0037 | 0.7019±0.0032 | 0.4931±0.0019 |
| SIM | 0.8313±0.0025 | 0.7103±0.0010 | 0.4902±0.0008 |
| PinSage | 0.8289±0.0036 | 0.7086±0.0031 | 0.4917±0.0017 |
| LightGCN | 0.8309±0.0006 | 0.7093±0.0018 | 0.4909±0.0012 |
| GLSM | 0.8326±0.0053 | 0.7149±0.0039 | 0.4887±0.0029 |
| GMT | 0.8343±0.0022 | 0.7178±0.0033 | 0.4862±0.0021 |
| **MacGNN** | **0.8408**±0.0019 | **0.7233**±0.0014 | **0.4817**±0.0013 |

**Table 4: Ablation study results between MacGNN with its four variants on MovieLens and Electronics.**

| | Variant | AUC (↑) | GAUC (↑) | Logloss (↓) |
|---|---|---|---|---|
| MovieLens | **MacGNN** | **0.7458**±0.0006 | **0.7198**±0.0007 | **0.5886**±0.0027 |
| | *w/o weighting* | 0.7396±0.0013 | 0.7132±0.0009 | 0.5923±0.0037 |
| | *w/o recent* | 0.7212±0.0005 | 0.6936±0.0009 | 0.6176±0.0052 |
| | *w/o highorder* | 0.7401±0.0004 | 0.7126±0.0009 | 0.5929±0.0030 |
| | *w/o itemgraph* | 0.7239±0.0002 | 0.6871±0.0007 | 0.6073±0.0032 |
| Electronics | **MacGNN** | **0.8444**±0.0009 | **0.8458**±0.0008 | **0.4892**±0.0040 |
| | *w/o weighting* | 0.8418±0.0006 | 0.8417±0.0006 | 0.4938±0.0032 |
| | *w/o recent* | 0.8316±0.0010 | 0.8333±0.0008 | 0.5127±0.0037 |
| | *w/o highorder* | 0.8302±0.0003 | 0.8319±0.0005 | 0.5083±0.0033 |
| | *w/o itemgraph* | 0.8189±0.0005 | 0.8199±0.0007 | 0.5259±0.0043 |

The comparison result is shown in Figure 5. From the figure, we have the following observations: (i) The proposed model is almost as efficient as the simplest Wide&Deep model. Apart from Wide&Deep, our model achieves the best performance and efficiency among all user interest-based models and sampling-based graph models. (ii) While graph models employ sampling strategies to expedite the inference process, LightGCN and GMT are the two slowest models. Particularly on online platforms, LightGCN and GMT require nearly three times and two times the inference time of MacGNN, leading to a significant online burden for billion-scale recommender systems.

## 4.4 Ablation Study (RQ3)

To verify the effectiveness of the key designed components and modeled information in MacGNN, we conduct the ablation study by comparing MacGNN with its four variants: (1) *w/o weighting* removes the *macro weight modeling* module in MacGNN, which ignores the macro edge weights. (2) *w/o recent* removes the *recent behavior modeling* scheme in MacGNN, of which the short-term pattern modeling. (3) *w/o highorder* excludes the high-order graph information of the target user and item for MacGNN training and

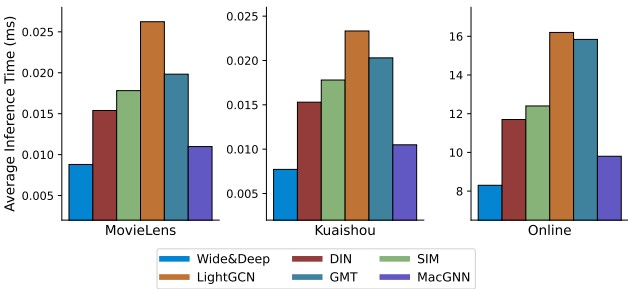

**Figure 5: Efficiency study of the model inference time.**

the final prediction. (4) *w/o itemgraph* excludes the target item's graph information for MacGNN training and prediction, which is largely ignored by previous works due to the efficiency trade-off. From Table 4., we have the following observations:

**Effectiveness of key designed components.** (i) The lack of consideration of the macro edge weight results in the inferior performance of *w/o weighting*, as the macro edge intensity can reflect the behavior pattern of users and items. (ii) The removal of recent behavior may impact the recommendation performance of *w/o recent* in comparison to MacGNN. This underscores the importance of taking recent behaviors into account from a macro perspective.

**Effectiveness of key modeled information.** (i) The decline in the performance of *w/o highorder* relative to MacGNN due to the neglect of high-order neighbors indicates the significance of graph information, and considering it from a macroscopic perspective is effective. (ii) The substantial performance gap between *w/o itemgraph* and MacGNN highlights the significance of considering item-side graphs. Nonetheless, traditional CTR models tend to discard them due to computational constraints.

## 4.5 Parameter Analysis (RQ4)

*4.5.1 **Effect of Temperature Parameter**.* We investigate the effect of the temperature parameter $\tau$ in macro node weighting with the range of 0.1 to 1.9 with a step size of 0.2 as illustrated in Figure 6. We can observe from the results that a too-small weighting value of $\tau$ will cause poor performance. Furthermore, the suitable value of $\tau$ for MovieLens is larger than 1 while for Electronics is smaller than 1, one possible reason is that the temperature of MacGNN should be set smaller on more sparse datasets.

*4.5.2 **Effect of Macro Node Number**.* We also evaluate the impact of different macro user numbers $\widetilde{n}$ under the behavior pattern grouping and fixed utilize of category as item grouping to avoid the impact of multiple variables. From the second row of Figure 6, we can find that the too-small cluster number will lead to too coarsen user segmentation and result in poor results. In addition, choosing a relatively appropriate number of clusters, such as 20, can bring good enough performance of MacGNN on the public datasets and this macro node number is much smaller than the micro interaction scale, and also much smaller than the sequence length of previous user interest modeling works [30, 31].

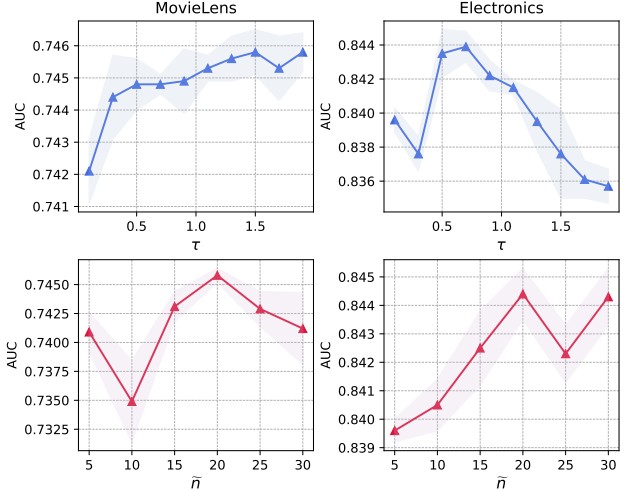

**Figure 6: Parameter study of temperature parameter $\tau$ and macro node number $\widetilde{n}$ on MovieLens and Electronics.**

## 4.6 Online Evaluation (RQ5)

We have deployed MacGNN and conducted the online A/B test in one of the biggest shopping platforms for over two months. The online performance is compared against the best performed user interest-based model SIM and the sampling-based graph model GMT. The performance in Table 5 is averaged over four consecutive weeks. We have the following observations.

*Compared to SIM*, firstly, MacGNN demonstrates a performance improvement of 3.13% for PCTR, 1.32% for UCTR, and 5.13% for GMV, suggesting that our model enhances users' willingness to engage with items and convert to purchases. Secondly, the Stay Time increases by 1.01%, indicating that MacGNN can effectively engage users, encouraging them to spend more time on the platform by catering to their comprehensive macro behavior interests. Thirdly, MacGNN achieves a Response Time that is 20.97% faster than SIM, showing that MacGNN achieves significantly improved performance and enhanced efficiency.

*Compared to GMT*, MacGNN still demonstrates a performance improvement of 2.35% for PCTR, 1.09% for UCTR, and 3.53% for GMV. This suggests that taking into account the complete macro behavior patterns of users and items can yield significantly better performance than considering only a small portion of sampled neighbors. Furthermore, the Stay Time increases by 0.69%, indicating that MacGNN encourages users to stay by considering more comprehensive behavior patterns. Lastly, MacGNN's Response Time is 38.13% faster than SIM, confirming the efficiency of MAG.

Both A/B testing results validate that MAG and MacGNN are more suitable than previous micro recommendation models.

**Table 5: Results of online A/B tests in the industrial platform.**

| A/B Test | PCTR | UCTR | GMV | StayTime | ResTime |
|----------|------|------|-----|----------|---------|
| v.s. SIM | +3.13% | +1.32% | +5.13% | +1.01% | -20.97% |
| v.s. GMT | +2.35% | +1.09% | +3.53% | +0.69% | -38.13% |

## 5 RELATED WORK

### 5.1 Click-Through Rate Prediction

Click-through rate (CTR) prediction is now central in online recommender systems [28]. Tradition models utilize feature interaction for CTR prediction. FM [19] first introduces the latent vectors for 2-order feature interaction to address the feature sparsity. Wide&Deep [2] conducts feature interaction by a wide linear regression model and a deep feed-forward network with joint training. DeepFM [9] further replaces the linear regression in Wide&Deep with FM to avoid feature engineering. Recently, user interest-based models have achieved better CTR performance. DIN [31] first designs a deep interest network with an attention mechanism between the user's behavior sequence and the target item. DIEN [30] then further enhances DIN with GRU [3] for user's evolution patterns mining. Similarly, SIM [17] designs a two-stage paradigm, searching relevant items and computing their attention score with the target, to reduce the scale of the user's complete behaviors.

Although successful, these models ignore the modeling of graph information due to the efficiency trade-off. It will lose some valuable information for precise interest modeling, which is also the motivation of the designed MAG and MacGNN.

### 5.2 Graph Learning for Recommendation

Recently, massive works have attempted to improve recommendation performance through graph learning methods [21, 24].

Typically, NGCF [23] enhances traditional collaborative filtering with high-order graph information. LightGCN [13] then removes the non-linear operation in NGCF, which is drawn from the observation of extensive experimental analysis. These methods have been widely used for appropriate item recalling in industrial recommender systems. However, due to the strict requirements for time efficiency, they cannot be applied directly as CTR prediction models. Then, some advances try to consider the graph information in the CTR scenario but they still maintain the node sampling paradigm. GLSM [20] conducts relevant node retrieval of the central user from the interaction graph for long-term interest modeling. GMT [15] constructs a heterogeneous information network (HIN) with sampled various types of user interactions and designs a graph-masked transformer for user modeling.

## 6 CONCLUSION

The introduction of the *Macro Recommendation Graph* and **Macro Graph Neural Networks (MacGNN)** has significantly advanced the field of billion-scale recommender systems, offering a viable solution to the prevalent issues of computational complexity and sampling bias in conventional GNN models. By ingeniously grouping micro nodes into macro nodes, MAG allows for efficient computation, while MacGNN facilitates effective information aggregation and embedding refinement at a macro level. Demonstrating superior performance in both offline experiments and online A/B tests, and practically serving over a billion users in a major shopping platform, this approach not only elevates the capability of predictive models in expansive digital environments but also paves the way for future research and optimizations in the realm of large-scale recommendation systems.

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

# A  DATASET DETAILS

We adopt both three publicly available datasets on a billion-scale industrial dataset for offline evaluation. The detailed description and preprocessing manner of the datasets are as follows:

**MovieLens Dataset**[2] [11] contains 71,567 users, 10,681 movies, and 10,000,054 interactions of users' ratings to the movies. To make the rating interactions suitable for the CTR prediction task, we follow the previous works [31] to transform the rating interactions into clicked and non-clicked relationships, which label the samples with rating values that greater than or equal to 4 to be positive and the rest to be negative.

**Electronics Dataset**[3] [14] is a subset of Amazon Dataset, which contains product reviews and metadata from Amazon. It contains 192,403 users, 63,001 items, and 1,689,188 interactions. We treat all the user reviews as user click behaviors, which is widely used in the related works [30, 31].

**Kuaishou Dataset**[4] [7] is a real-world dataset collected from the recommendation logs of the video-sharing mobile app Kuaishou. It contains 7,176 users, 10,728 videos, and 12,530,806 interactions. We regard the samples with video play time account for more than 50% of the total time to be truly clicked videos.

**Industrial Dataset** is a large-scale dataset collected from one of the largest e-commerce recommendation applications, involving billions scale of users and items. The industrial dataset contains both positive and negative interactions (e.g., impression without user clicks) such that negative sampling is not needed. There are over 118 billion instances and each user has around 938 recent behaviors on average, which is much longer than the sequences from the public dataset. Following SIM [17], we use the instances of the past two weeks as the training set and the instances of the next day as the test set. The number of macro user clusters is 200, while the number of macro item clusters is 300.

# B  BASELINE DETAILS

We compare our proposed MacGNN with twelve representative state-of-the-art CTR prediction models as follows.

**Feature Interaction-based Methods:** (i) **Wide&Deep** [2] is widely used in real industrial applications. It consists of a wide module and a deep module to discover and extract the correlation and nonlinear relations between features. (ii) **DeepFM** [9] is a variant model of Wide&Deep, which imposes a factorization machine (FM) [19] as a wide part avoiding manufactured feature engineering. (iii) **AFM** [25] improves feature interactions by discriminating the different importance via an attention network. (iv) **NFM** [12] introduces the bi-interaction pooling to deepen FM for learning higher-order and non-linear feature interactions.

**User Interest Modeling-based Methods:** (i) **DIN** [31] is the first model that uses an attention mechanism to extract user interest representation from truncated historical user behaviors in CTR prediction. (ii) **DIEN** [30] is an improved version of DIN, which uses a two-layer RNNs module enhanced with the attention mechanism to capture the evolving user interests. (iii) **UBR4CTR** [18] proposes a search engine-based method to retrieve more relevant

---

[2]https://grouplens.org/datasets/movielens/10m
[3]https://jmcauley.ucsd.edu/data/amazon
[4]https://kuairec.com

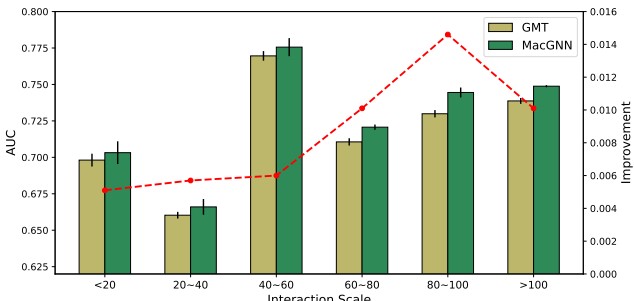

Figure 7: Case study of user groups with different interaction scales on MovieLens dataset.

and appropriate behavioral data in long user sequential behaviors for model training. (iv) **SIM** [17] uses two cascaded search units to extract user interests, which has a better ability to model long sequential behavior data in both scalability and performance in the CTR prediction.

**Graph-based Methods:** (i) **PinSage** [26] is a representative graph-based web-scale recommendation model, which conducts inductive graph aggregation on the sampled user/item nodes. We concatenate and feed the trained embeddings by PinSage into the widely employed prediction layer to fit the CTR prediction scenario. (ii) **LightGCN** [13] is a simplified collaborative filtering model design by including only the most essential components in GCN for recommendation. Since it is a collaborative filtering model, the trained embeddings are also fed into the prediction layer as PinSage for the CTR prediction. (iii) **GLSM** [20] is a sampling-based model to introduce graph information, which consists of a multi-interest graph structure for capturing the long-term patterns and a sequence model for modeling the short-term information. (iv) **GMT** [15] is also a sampling-based state-of-the-art graph model for CTR prediction with a graph-masked transformer to learn different kinds of interactions on the heterogeneous information network among the constructed neighborhood nodes.

# C  CASE STUDY

We further conduct the case study to verify the performance of MacGNN on users with different interaction frequencies. Specifically, we divided users into 6 groups according to their interaction frequency on the MovieLens dataset. The case study results are illustrated in Figure 7.

We can find that our MacGNN performs better in most cases, which shows that the introduction of MAG can benefit users with different interaction frequencies. This observation can be explained in the following two main aspects: (i) For low-active users, the modeling view from a macro perspective will bring additional general key features, and the high-order graph information from MAG also provides helpful information for user modeling. (ii) For high-active users, in addition to ensuring computational efficiency, macro modeling on MAG can also avoid noise and overly complex information contained in excessively long interaction sequences. Thus, besides improving computational efficiency for considering both complete and high-order patterns, the organization of MAG is also beneficial for modeling interests in various interaction frequencies.

