# OpenReview forum: "Online Billion-Scale Recommender Systems with Macro Graph Neural Networks"
_ACM.org/TheWebConf/2024/Conference — TheWebConf24_

### Official Review · Reviewer_jSpn · 2023-11-20

**Novelty:** 5
**Technical Quality:** 6

**Review:**

### Paper Summary
This work proposed a novel macro node (clusters) of user/item nodes in the 1-hop and 2-hop neighbors of user-item graph, with the benefits of lower computational complexity and better ctr performances. The experiments are conducted in both open datasets and one industrial-level dataset as an online a/b test.

### Clarity
The overall workflow can be easily understood by the figure. However, the equations are chaotic. In equation 5, what is and how to calculate $r_{ab}$? Is that a rating score? What is the dimension of $\textbf{b}_v$? The L2-norm output is a scalar instead of an embedding vector.
Is the workflow from equations 2 to 4 just the kmeans clustering? Is there any popularity bias introduced in the clustering stage if the calculation is on the raw interaction vector?

### Originality
I believe the idea of this work sounds novel and interesting. It is pretty interesting to see a work utilizing the cluster structure for both better efficiency and accuracy.

### Significance
This work can benefit both the research and industrial communities for better efficiency and accuracy for ctr prediction. This can also inspire the research community to realize the importance of using cluster or inherent structures of users and items in personalized recommendation.

**Pros**
The idea of this work sounds novel and interesting, and will have beneficial impact to the community.

**Cons**
The equations are chaotic and sometimes difficult to understand.

**Questions:**

* Please help clarify the unclear points in the clarity section above.
* As both modeling modules in equations 8 - 11 are just self-attention mechanism, why don't authors also compare the transformer approach? such as pinformer, autoint, or regular transformer?

**Reviewer Confidence:**

4: The reviewer is certain that the evaluation is correct and very familiar with the relevant literature

**Scope:**

4: The work is relevant to the Web and to the track, and is of broad interest to the community

---

### Official Review · Reviewer_2fdi · 2023-11-22

**Novelty:** 6
**Technical Quality:** 6

**Review:**

To take more useful neighbor information from user-item interactions, this paper propose a Macro Graph Neural Network-based approach. Different from sequential user-behavior modeling, such as DIN and SIM, this work is able to utilize high-order relationship between users and items. Compared to traditional GNN-based user behavior modeling, this approach alleviates the sampling bias problem and captures the full spectrum of user or item behavioral pattern. Both online and offline experiments are conducted.

**Questions:**

From the section METHODOLOGY, the macro nodes are generated by grouping the micro nodes' embedding? An how to define the number of macro nodes?

Based on the generated macro nodes, how many kinds of methods can construct the relationship between these nodes? Pls. give a detailed discussion. This part is interesting.

In Figure-2, only 2-Hop neighbors are utilized, what is the highest order of neighbors which is helpful for final performance.

How to treat and assign the macro nodes for the new items and users?

**Reviewer Confidence:**

4: The reviewer is certain that the evaluation is correct and very familiar with the relevant literature

**Scope:**

4: The work is relevant to the Web and to the track, and is of broad interest to the community

---

### Official Review · Reviewer_XXZn · 2023-11-24

**Novelty:** 4
**Technical Quality:** 4

**Review:**

The paper introduces an approach to billion-scale recommender systems by proposing a Macro Recommendation Graph (MAG) and a corresponding Macro Graph Neural Network (MacGNN). The key contributions of the paper can be summarized as follows:

MAG involves the construction of macro nodes, macro edges, and macro subgraphs. This customization reduces the number of neighbors from billions to hundreds, eliminating the need for sampling strategies and facilitating Graph Neural Network (GNN) operations in billion-scale recommender systems.

MacGNN is introduced as a novel paradigm for efficient CTR prediction in billion-scale recommender systems. It aggregates macro-graph information and updates macro-node embeddings, providing a solution to the challenges faced by traditional GNNs in handling large-scale neighbor complexities.

Extensive offline experiments conducted on three public benchmark datasets and a billion-scale industrial dataset show the performance of MacGNN compared to state-of-the-art CTR baselines.
Online A/B tests further confirm the performance of MacGNN in real-world billion-scale recommender systems.


While the paper introduces a promising approach to address challenges in billion-scale recommender systems through MAG and MacGNN, it is essential to acknowledge certain limitations and concerns, including the insufficiency of code details for result reproduction.
The paper may fall short in providing sufficient code details and implementation specifics for reproducing the results. The absence of a comprehensive codebase or clear guidelines might hinder researchers or practitioners from replicating the experiments, potentially impacting the credibility and transparency of the proposed approach.

**Questions:**

Why have you chosen that clustering strategy that resembles k-means? Have you tried other options?

Could you better detail how macro-edges are computed?

**Ethics Review Description:**

No issues

**Reviewer Confidence:**

3: The reviewer is confident but not certain that the evaluation is correct

**Scope:**

4: The work is relevant to the Web and to the track, and is of broad interest to the community

---

### Official Review · Reviewer_srJW · 2023-11-25

**Novelty:** 3
**Technical Quality:** 4

**Review:**

In this paper, the authors propose a macro recommendation graph for online billion-scale recommendations. Reducing the complexity of graph neural networks is definitively an essential problem.

One unclear part is the design choices for constructing nodes and edges in the graph. I consider the main idea of the paper is to group users and items into several groups to reduce the size of the graph. Therefore, I hope the authors clearly compare the design choices of the grouping.

**Questions:**

Please correct me if I have some misunderstanding parts in the review.

**Reviewer Confidence:**

4: The reviewer is certain that the evaluation is correct and very familiar with the relevant literature

**Scope:**

4: The work is relevant to the Web and to the track, and is of broad interest to the community

---

### Official Review · Reviewer_Jd2x · 2023-12-01

**Novelty:** 6
**Technical Quality:** 6

**Review:**

The paper titled "Online Billion-Scale Recommender Systems with Macro Graph Neural Networks" addresses the challenge of predicting Click-Through Rate (CTR) in billion-scale recommender systems, a pressing issue in platforms with vast numbers of users, items, and interactions. Conventional Graph Neural Networks (GNNs) struggle with the computational complexity involved in aggregating information from billions of neighbors, often resorting to sampling a small portion of neighbors which leads to severe sampling bias and fails to encompass the full spectrum of user or item behavioral patterns. Authors introduce MacGNN by aggregating nodes into Macro nodes, which effectively reduces the neighbor number and efficiency of graph-based CTR prediction. Especially, the algorithm is further tested in online systems, which is another good point for this paper.

Strength:
1. The macro node idea is applicable and interesting, which greatly reduces the computation cost.
2. This paper discerns the neighborhood distribution of user/item, which is overlooked by previous research.
3. The online testing makes this paper unique among all the other candidates.

**Questions:**

This paper is novel, solid, and interesting. I do not have further questions.

**Reviewer Confidence:**

3: The reviewer is confident but not certain that the evaluation is correct

**Scope:**

4: The work is relevant to the Web and to the track, and is of broad interest to the community

---

### Decision · Program_Chairs · 2024-01-22

**Decision:**

Accept

**Comment:**

By summarizing the review comments and responses, the ideas of this paper are applicable and interesting, and the authors also did extensive experiments to prove the effectiveness of their proposed method. However, one reviewer has concerns about how to construct graphs. And there are some details about the method that need to be clarified. I recommend that the authors should fix all issues in their camera-ready version.